# The association between racism and psychosis: An umbrella review

India Francis-Crossley[1]*, Georgie Hudson[1], Lasana Harris[2], Juliana Onwumere[3,4,5☙], James B. Kirkbride[1☙]

1 Division of Psychiatry, University College London, London, United Kingdom, 2 Department of Experimental Psychology, University College London, London, United Kingdom, 3 Department of Psychology, Institute of Psychiatry, Psychology and Neuroscience, King's College London, London, United Kingdom, 4 South London and Maudsley NHS Foundation Trust, London, United Kingdom, 5 NIHR Biomedical Research Centre for Mental Health South London and Maudsley NHS, London, United Kingdom

☙ These authors contributed equally to this work.
* india.francis-crossley.22@ucl.ac.uk

## Abstract

Elevated rates of psychosis are consistently identified in people from racialised backgrounds, with growing evidence from the systematic review literature that suggests a role for racial/ethnic discrimination. We synthesised current systematic review evidence on the association between racial/ethnic discrimination and psychosis. We conducted an umbrella review, systematically searching Medline, Embase, PsycINFO, ProQuest Central and Google Scholar (up to 19 July 2024) for systematic reviews and meta-analyses published in peer-reviewed journals exploring the effect of racial/ethnic discrimination on psychosis. 2898 citations were de-duplicated and screened, included reviews were assessed for risk of bias using AMSTAR-2 and extracted data analysed narratively following a pre-registered protocol (CRD42023400656). Seven reviews (reporting 23 primary studies representing 40,300 participants) met inclusion criteria, five of which explicitly reported on the association between racial/ethnic discrimination and psychosis. All observed evidence of a positive relationship between the two, including meta-analyses for psychotic symptoms (adjusted OR=1.77, 95%CI 1.26, 2.49) and psychotic experiences (pooled OR=1.94, 95%CI 1.42, 2.67). We observed more robust evidence for psychotic outcomes in non-clinical populations. Despite this, results were driven by cross-sectional studies (87%) and were hindered by high heterogeneity and low (n=2) or critically low (n=5) AMSTAR-2 review quality scores. The available systematic review evidence supports a role for racial/ethnic discrimination in developing psychosis, but high-quality studies are needed to determine the temporal and mechanistic causal pathways through which this occurs. The current findings add to knowledge on the widespread presence and deleterious impacts of racism on health and

**Data availability statement:** The data collected and used as part of this umbrella review (i.e. the data extracted from included reviews and the analysed data) are available at the following DOI: 10.17605/OSF.IO/GX3DQ

**Funding:** This project was funded by the UCL-Windsor Fellowship Research Opportunities scholarship (https://www.windsor-fellowship.org/; https://www.ucl.ac.uk/) (to IFC), a Wellcome Trust PhD Fellowship in Mental Health Science (218497/Z/19/Z) (https://wellcome.org/) (to GH), the Mental Health Mission Early Psychosis Workstream (NIHR203316) (www.nihr.ac.uk) (to JBK) and by UK Research and Innovation (UKRI) funding for the Population Mental Health Consortium (Grant no MR/Y030788/1) which is part of Population Health Improvement UK (PHI-UK) (https://www.phiuk.org/), a national research network which works to transform health and reduce inequalities through change at the population level (to JBK). The funders had no role in study design, data collection and analysis, decision to publish, or preparation of the manuscript. JO is part supported by Wellcome [308556/Z/23/Z] and the National Institute for Health Research's (NIHR) Biomedical Research Centre at South London and Maudsley NHS Foundation Trust and King's College London. This paper represents independent research funded by the Wellcome Trust [308556/Z/23/Z]. The funders had no involvement in study design, data collection, analysis, interpretation or the decision to submit for publication. The views expressed are those of the author(s) and not necessarily those of the funders.

**Competing interests:** The authors have declared that no competing interests exist.

inform potential public health interventions that reduce exposure to, and the impact of, racial/ethnic discrimination.

## Introduction

Psychosis is a severe mental health condition that has detrimental impacts on people's lives. For example, schizophrenia accounts for 12.2% of disability-adjusted life-years due to mental conditions and is ranked as the 20th leading cause of years lived with disability globally [1]. Further, people living with schizophrenia are estimated to lose an average of 14.5 years of potential life [2], due to both increased suicide risk (occurring in approximately 15% of cases [3]) and higher prevalence of comorbid physical health conditions, including cerebrovascular and cardiovascular diseases [4]. People living with psychosis also experience high rates of unemployment [3], barriers to accessing physical healthcare [5], and continue to be exposed to elevated levels of stigma and social exclusion [6]. These factors can further extend their health and social inequalities.

Psychotic disorders are not distributed equitably within populations. Longstanding ethnic disparities in psychosis risk are well-documented, particularly in Europe, Canada, Australia and the United States (US) [7–10]. For example, evidence from England shows that incidence rates of psychotic disorder are 2–5 times higher for people from Bangladeshi, Pakistani and Black ethnic backgrounds compared with White British groups [11]. A recent meta-analysis conducted in the US also showed that people from Black ethnic backgrounds were two times as likely to be diagnosed with a psychotic disorder than White individuals [10], with another estimate suggesting that people from Latine and Black ethnicities were 2 and 4 times more likely to experience psychosis, respectively, than people from White backgrounds [12].

Various factors have been proposed to explain elevated psychosis rates in people from racialised backgrounds [13], with differing levels of empirical support. These include genetic variation in psychosis risk by ethnicity (for which there is little evidence [14], particularly given 'race' and ethnicity are social constructs), selective migration, and exposure to various social adversities that causally increase risk. Clinician bias/misdiagnosis has also been long proposed as an explanation [15–18], however, since elevated psychosis risk has also been observed at both clinical and non-clinical ends of the psychosis spectrum, clinical misdiagnosis alone cannot explain these systematic increases [19].

Recently, various social determinants of health [19–21] have been implicated in our understanding of racial/ethnic disparities in psychosis risk. These factors include but are not limited to exposure to lower socioeconomic positions and adverse neighbourhood conditions, including deprivation and social exclusion; experience of traumatic events, including direct experiences of racism, persecution, food insecurity, violence or war, and; greater experience of early life obstetric complications amongst many racialised groups [21–23]. Many of these factors may be broadly conceptualised under the umbrella term, systemic racism – one of many forms of racism.

Racism itself can be broadly defined as the unequal distribution of power, resources and/or opportunities due to a person's ethnicity or race. The act or experience of racism is defined as racial/ethnic discrimination and includes behaviours such as exclusion, attacks and microaggressions. Racism can take many different forms, including interpersonal, internalised, vicarious and structural/institutional racism [20,24,25], and can therefore be pervasive in the ways in which it impacts people from racialised backgrounds, from personal through to systemic levels of oppression. In fact, the impact of such racial/ethnic discrimination may also be passed on to later generations as generational trauma through epigenetic factors, such as in the descendants of enslaved peoples [26].

There is emerging evidence for a role of racial/ethnic discrimination as a potential risk factor for ethnic disparities in psychosis risk [11,14,20,27–37]. Nonetheless, several recent systematic reviews appear to present a disparate picture of a methodologically variable literature. To our knowledge, no umbrella review has been conducted to synthesise and understand the totality of evidence on this issue.

This umbrella review sought to synthesise the current systematic review evidence on the association between racial/ethnic discrimination and psychosis. We aimed to answer the following research questions: 1) To what extent is racial/ethnic discrimination associated with psychosis risk in people from racialised backgrounds? 2) Do these associations vary by: type of racial/ethnic discrimination, clinical or non-clinical sample, racial/ethnic background, country, or study design? We hypothesised there would be strong evidence for a positive association between more experiences of racism and increased psychosis risk, and that this relationship would be similar by type of racism, sample, ethnic background, country or study design.

## Methodology

### Overview

Our umbrella review (a systematic review of reviews) followed the Preferred Reporting Items for Systematic Reviews and Meta-Analyses (PRISMA) guidelines [38] (S1 Checklist) and Cochrane overview of reviews handbook guidelines [39]. We pre-registered the protocol on PROSPERO (ID: CRD42023400656).

### Terminology

We generally followed the nomenclature used in the original reviews when reporting effects in specific racial/ethnic groups (i.e., Black, African-American, Asian etc.). Exceptions to this included our use of the term "Latine", as a non-gendered term in the place of latino/a/x, and our use of "racialised background" to refer to people who identified with, or were reported as being from a marginalised racial or ethnic group. We used such terminology as an alternative to potential outdated terms such as BAME/BME or "Blacks" as these are exclusionary, can be dehumanising, and focus more on socially-constructed categories rather than the experiences people from racialised backgrounds face. Similarly, when describing experiences of racism, we referred to "racial/ethnic discrimination" to cover the full breadth of experiences reported by the included reviews.

### Search strategy

We searched Medline, Embase, PsycINFO and ProQuest Central with no date restrictions (up to 19 July 2024), using search terms related to 1) racial/ethnic discrimination (i.e., racism, ethnic bias, perceived discrimination and prejudice), 2) psychosis (i.e., schizophrenia, psychotic disorder, paranoia, psychosis), and 3) review (systematic reviews and meta-analyses). Additionally, we searched Google Scholar using an adapted version of the search concepts, using the following search terms: 1) racism/racial discrimination, 2) psychosis/schizophrenia, and 3) systematic review/meta-analysis (further details in S1 Text). We also conducted forward- and backward-citation searching (searching for reviews included in the systematic review reference lists, and reviews that cited the included systematic

reviews) of included systematic reviews to identify any potential reviews missed by our initial searches. The original search was conducted in March 2023, and re-run in July 2024 to cover all published papers until 19 July 2024. Our full search strategy is provided in S1 Text.

### Inclusion criteria

Our criteria for inclusion were:

1) Systematic reviews and/or meta-analyses

2) Investigation of *non-organic psychosis* (including psychotic disorders, psychotic symptoms psychotic-like experiences, at-risk mental states or ultra-high risk (UHR) for psychosis) and *racial/ethnic discrimination* (i.e., discrimination based on race, ethnicity or ethnocultural background)

3) Include people from racialised backgrounds

4) Published in a peer-reviewed journal.

Reviews with a wider scope than this umbrella review (e.g., multiple health outcomes, not limited to psychosis) were eligible for inclusion as long as they planned to search/include psychosis. Our definition of psychosis included the full continuum from psychotic-like experiences through to psychotic symptoms and clinically diagnosed psychotic disorders. For the purpose of this review we defined psychotic experiences and symptoms as disturbances in perception, thought or belief, such as delusions and hallucinations, and differentiated between reviews and original studies that reported these in non-clinical (i.e., population-based) and clinical (i.e., help-seeking) samples.

We placed no restrictions on participants by age, gender or country. We excluded non-systematic reviews (e.g., narrative and scoping reviews), the grey literature, primary studies, and citations where racial/ethnic discrimination was not reported separately from other forms of discrimination, or where reporting of racial/ethnic group was restricted to binary classifications of racialised groups (i.e., all Black and Minority Ethnic groups often abbreviated to 'BME', 'BAME' or 'non-white').

### Data management and extraction

We managed search results and de-duplication using Endnote (version 20) (Clarivate, London, UK). Title, abstract and full-text screening, data extraction and risk of bias assessments, were carried out in duplicate at each stage (IFC and GH) using Covidence systematic review software (Veritas Health Innovation, Melbourne, Australia), following training on the first 30% (n = 570) of the initial title and abstract screening records to ensure high inter-rater reliability (IRR), measured using Cohen's *Kappa* at each stage ($K \geq 0.80$ indicate good agreement [40]). Disagreements were resolved through consensus discussion to reach agreement (between IFC and GH), with any unresolved disagreements agreed in conjunction with a senior author (JBK).

We extracted the following descriptive characteristics from included reviews:

- Relevant included primary studies and their study and population characteristics (e.g., year of publication, study design, total number of participants, participant ages, sex/gender, and racial/ethnic background);

- Review search strategies;

- Review exposure and outcomes variables;

- Review results;

- Funding source(s) and conflict(s) of interest.

During data extraction we differentiated studies by psychosis outcome type, distinguishing between those conducted in clinical and/or non-clinical samples. Within clinical samples, we further distinguished between studies of risk/incidence/ diagnostic category of psychotic disorder and studies of severity of psychotic symptoms in ultra-high risk (UHR) and clinically-diagnosed samples. Within non-clinical samples, we identified studies of psychotic-like experiences and psychotic symptoms in the general population, as reported by the reviews.

We categorised type of racial/ethnic discrimination according to forms of racism that met our definition (see introduction): interpersonal, internalised, vicarious and structural/institutional racism. We also extracted data according to other types of racism, where these were specified within the reviews or primary studies (e.g., work-related racial/ethnic discrimination).

In accordance with the Cochrane handbook [39], we extracted data as reported by the reviews themselves, without returning to the relevant primary studies to extract data. The exception to this was in cases where two or more reviews had reported discordant data from the same primary study. Where any included systematic review did not explicitly report primary study data on racial/ethnic discrimination and psychosis (despite an intention to include such primary studies, should they have existed), we contacted the corresponding author of that review to clarify this, and provide data where it existed.

### Risk of bias

We estimated risk of bias of included reviews using the 16-item AMSTAR-2 checklist [41]. The checklist includes seven critical domains (items 2, 4, 7, 9, 11, 13 and 15; see S1 Table). We followed AMSTAR-2 guidelines to rate reviews as of high, moderate, low or critically low quality, according to prespecified adherence to the critical and non-critical criteria (S1 Table). Additionally, we extracted risk of bias scores of the primary studies where reported by the systematic reviews (S2 Table), in alignment with AMSTAR-2 item 9. As AMSTAR-2 has been criticised for low scoring when applied to observational studies [42], we also calculated the number of items each review met (completely or partially) to provide further detail.

### Data synthesis

We planned to perform meta-analyses where there were commonalities in how the effect sizes between psychosis and racial/ethnic discrimination were measured in at least five primary studies, and where both the effect size and a measure of standard error were reported or could be estimated. If there were insufficient data to do this, we reported any meta-analyses conducted in the included reviews. Additionally, we conducted a narrative synthesis of results by: outcome type (see above); exposure (type of racial/ethnic discrimination); and primary study characteristics (study design, country of study, and racial/ethnic background). We reported comparator groups as defined by the included reviews. Where characteristics (e.g., type/frequency/administration) of the instrument(s) used to assess the exposure and outcome variables were not reported by the reviews, we inferred them based on the original development of each reported measure/scale.

We used a citation matrix to identify relevant primary studies reported by multiple systematic reviews, and calculated the 'corrected covered area' using methodology from Pieper et al. [43] to measure the extent of overlap (0–100%) across included systematic reviews.

### Publication bias

We reported publication bias as assessed by the included reviews. Where we were able to synthesise data from meta-analyses, small study effects were to be assessed using Egger's test [44].

### Deviations from protocol

We made four deviations from the protocol due to methods that emerged as redundant during the review process: 1) amendment to exclude non-systematic narrative reviews, in line with Cochrane guidance [39]; 2) removal of SWiM [45]

reporting guidelines for narrative synthesis, as these are intervention-focused; 3) removal of GRADE [46] appraisal, as we included a separate quality/risk of bias assessment using AMSTAR-2; and 4) removal of the English language screening restriction.

## Results

### Search and screening

Our initial search identified 2601 records (Fig 1), of which 26.9% (n = 700) were duplicates. The remaining 1901 records were screened, 86.6% (n = 1647) of which were excluded at title screen (IRR = 0.85), 9.9% (n = 188) at abstract screen (IRR = 0.83) and 3.1% (n = 59) at full text screen (IRR = 0.82). We were unable to obtain the full text for two records despite exhaustive searches, leaving five reviews (0.3%) that met inclusion criteria. We identified two additional reviews that met inclusion criteria during forward- and backward-citation screening, resulting in seven reviews included in this umbrella review [32,33,47–51].

### Review characteristics

All seven reviews were published between 2003 and 2023 (Table 1), of which four (57%) included meta-analyses. Four reviews (57%) included risk of bias assessments; two used the authors' own measure [48,49], one used the Effective Public Health Practice Project (EPHPP) tool [50], and one used the Strengthening the Reporting of Observational Studies

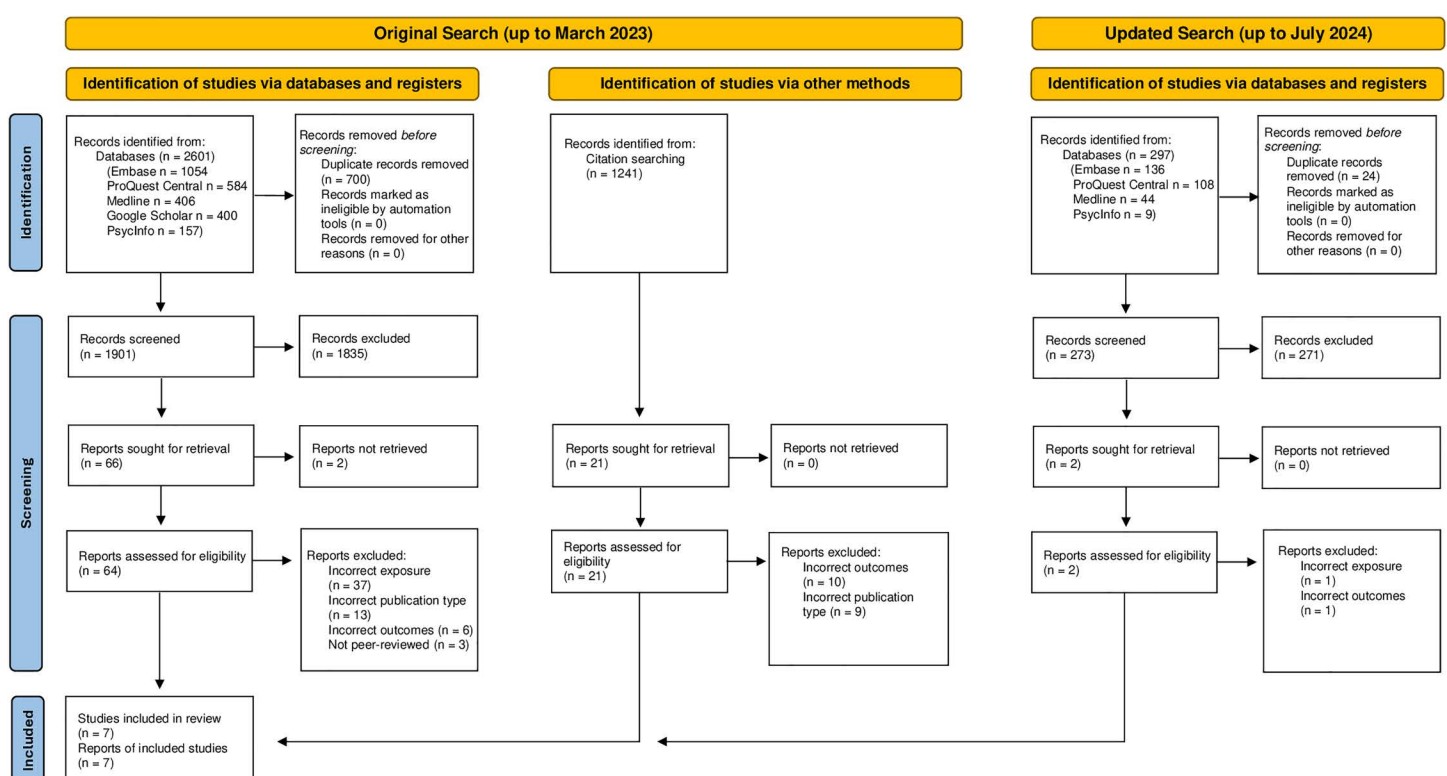

**Fig 1. PRISMA diagram to identify reviews included in this umbrella review.** PRISMA diagram of search and screening results. The results displayed for the original search (conducted in March 2023), also include the non-English language results which were re-screened in parallel with the updated screen (conducted in July 2024) due to the updated methodology to remove the English Language only restriction.2.

**Table 1. Main characteristics of the reviews included in our umbrella review.**

| Review | Year of publication | Title | Study aim(s) | Type of Review | Total primary studies | Number of relevant primary studies[1] | Risk of bias tool used | Relevant exposure variable(s) | Relevant outcome variable(s) | Comparator(s) | Databases searched | Funding source[3] | Conflicts of interest[3] |
|---|---|---|---|---|---|---|---|---|---|---|---|---|---|
| Bardol et al [47] | 2020 | Perceived ethnic discrimination as a risk factor for psychotic symptoms: a systematic review and meta-analysis | To conduct a systematic literature review and meta-analysis investigating the association between PED and PS/PE in people from ethnic minorities | Systematic Review & Meta-analysis | 18[2] | 18[2] | STROBE checklist | Perceived ethnic discrimination | Psychosis symptoms/experiences | General population | Medline, PsycINFO, Web Of Science | NR | NR |
| de Freitas [48] | 2018 | Psychological Correlates of Perceived Ethnic Discrimination in Europe: A Meta-Analysis | To conduct a systematic review and meta-analysis investigating the associations between perceived ethnic discrimination and psychological functioning (including psychosis) in people living in European countries | Systematic Review & Meta-analysis | 51 | 2 | Author's own measure | Perception of ethnic discrimination | Manifestation of symptoms of psychosis | NR | PsycINFO, PsycARTICLES, Psychology and Behavioral Sciences Collection, Education Research Complete, ERIC, Medline, SocIN-DEX, ProQuest Dissertations & Theses | Reported | NR |
| Paradies et al [49] | 2015 | Racism as a Determinant of Health: A Systematic Review and Meta-Analysis | To conduct a systematic review and meta-analysis on the association between reported racism and health outcomes (mental and physical) | Systematic Review & Meta-analysis | 333 | 6 | Author's own measure | Reported racism | Other mental health symptoms (e.g., paranoia, psychoticism)[4] | NR | Medline, PsycInfo, Sociological Abstracts, Social Work Abstracts, ERIC, CINAHL, Academic Search Premier, Web of Science, ProQuest (for dissertations/theses). The authors also identified additional reports from their personal databases and the reference lists of 25 major literature reviews, meta-analyses and other relevant works | Reported | The authors declared no conflict of interest |

*(Continued)*

**Table 1.** (Continued)

| Review | Year of publication | Title | Study aim(s) | Type of Review | Total primary studies | Number of relevant primary studies[1] | Risk of bias tool used | Relevant exposure variable(s) | Relevant outcome variable(s) | Comparator(s) | Databases searched | Funding source[3] | Conflicts of interest[3] |
|---|---|---|---|---|---|---|---|---|---|---|---|---|---|
| Pearce et al [50] | 2019 | Perceived discrimination and psychosis: a systematic review of the literature | To conduct a systematic review of quantitative cross-sectional and prospective studies examining the association between discrimination and psychosis in people from minority groups | Systematic Review | NR | 16 | EPHPP | Perceived racial discrimination | Psychosis/ psychosis symptoms | Controls (i.e., participants who have not experienced psychosis or psychotic experiences) | PsycINFO, Embase, PubMed | NR | The authors declared no conflict of interest |
| Pieterse et al [51] | 2012 | Perceived Racism and Mental Health Among Black American Adults: A Meta-Analytic Review | To conduct a meta-analysis into the effect size of the relationship between racism and mental health for Black Americans | Systematic Review & Meta-analysis | NR | 0 | Not assessed/ reported | Perceived Racism | Psychiatric Symptoms[5] | NR | Databases searched included PsycINFO, MEDLINE, Social Sciences Abstracts, CINAHL. The authors also identified reports from previously published reviews including Carter, 2007; Paradies, 2006; and Williams & Williams-Morris, 2000. | NR | NR |
| Williams and Mohammed [33] | 2009 | Discrimination and racial disparities in health: evidence and needed research | To conduct a review of the literature published on PubMed between 2005 and 2007 exploring the effect of racism on health updating the findings by the Paradies, 2006 review (2000–2004) | Systematic Review | 115 | 2 | Not assessed/ reported | Racial Discrimination | Psychosis | NR | PubMed | Reported | NR |

(Continued)

**Table 1.** (Continued)

| Review | Year of publication | Title | Study aim(s) | Type of Review | Total primary studies | Number of relevant primary studies[1] | Risk of bias tool used | Relevant exposure variable(s) | Relevant outcome variable(s) | Comparator(s) | Databases searched | Funding source[3] | Conflicts of interest[3] |
|---|---|---|---|---|---|---|---|---|---|---|---|---|---|
| Williams et al [32] | 2003 | Racial/Ethnic Discrimination and Health: Findings From Community Studies | To update reviews previously conducted (in 1999 and 2000) into the relationship between racial/ethnic discrimination and health in population-based studies | Systematic Review | NR | 1 | Not assessed/reported | Perceived racial/ethnic discrimination | Psychosis | NR | Medline, PsychINFO, Sociofile | Reported | NR |

[1]Number of all primary studies included in the review that met our inclusion criteria.

[2]Bardol et al. [47] included 17 primary citations, of which one, Kong [52], included two relevant primary studies, leading to 18 primary studies in this review.

[3]Funding source and conflict(s) of interest are reported in line with AMSTAR-2 guidelines.

[4]These results include irrelevant studies reported within the 'Other mental health symptoms (e.g., paranoia, psychoticism)' outcome.

[5]Up to 12 studies were included in the 'Psychiatric Symptoms' outcome, however, we were unable to obtain psychosis data from this category, and it is unlikely that any relevant studies for this umbrella review are included.

CI = Confidence interval; MH = mental health; PE = Psychotic experiences; PED = Perceived ethnic discrimination; PS = Psychotic symptoms; NR = Not reported.

in Epidemiology (STROBE) checklist [47]. Using the AMSTAR-2 appraisal, we assessed reviews as low (n = 2, 29%) [47,50] or critically low (n = 5, 71%) [32,33,48,49,51] quality (S1 Table). This was driven by: no prior registration of reviews (n = 5, 71%) [32,33,48,50,51]; statistical combination methods not meeting criteria recommended in the AMSTAR-2 guidelines (n = 2) [49,51]; not accounting for individual studies' risk-of-bias in the review interpretation or discussion of results (n = 4, 57%) [32,33,48,51], and; no or unsatisfactory risk of bias assessments for included primary studies (n = 6, 86%) [32,33,47–49,51] (including the review which used STROBE [47], which is considered unsatisfactory according to AMSTAR-2 guidelines). The percentage of total AMSTAR-2 items met by each review ranged from 6% [33] to 75% [47] (S1 Table), with three reviews [47,49,50] meeting 50% or more.

The identified reviews reported heterogenous measures and definitions of psychosis risk and racial/ethnic discrimination (Table 2), meaning we were unable to perform meta-analyses in the present umbrella review.

## Primary study characteristics

The seven included reviews reported 516 total papers (before de-duplication) of which 22 papers [27,29–31,52–69] reported on 23 primary studies (Kong [52] reported two relevant studies) containing original data on racial/ethnic discrimination and psychosis. Four reviews did not report or disaggregate primary study characteristics or results for psychosis alone, though we were able to obtain this disaggregated data from the authors of three of these four reviews [48–50]. The author of the fourth review [51] could not provide further relevant data; based on information from the review, we believe it did not identify any relevant primary studies on psychosis and racial/ethnic discrimination.

The overlap in the 23 primary studies reported by the reviews was very high ('corrected covered area' = 15.9%; Fig 2A). This overlap reduced the amount of missing primary study participant characteristics where under-reported in some reviews. Nevertheless, review reporting of some characteristics, such as participant age and gender, remained low at 52% and 57%, respectively (Fig 2B). The 23 studies were published between 1999–2023 and were conducted in the US (n = 8, 35%), United Kingdom (UK) (n = 7, 30%), The Netherlands (n = 5, 22%), Norway (n = 1, 4%), or Romania (n = 1, 4%) (Table 2); the location for one study was not reported [53]. Most study designs were cross-sectional (n = 20, 87%), while only two [52,54] (9%) were longitudinal.

## Participant characteristics

Four reviews [33,47,49,50] reported a total of 40,300 participants from the 22 primary studies they represented (Table 3); no review reported the sample size from the remaining relevant primary study [58]. Only one review [50] reported mean ages of participants, available from 12 of the 16 primary studies it included; mean ages ranged from 19.9 to 44.0 years, with a median of 37.9 years. Only one review [50], representing 13 of the 16 primary studies it included, reported on participant gender balance (n = 10,209 female (54.7%), n = 8,459 male (45.3%)). Three reviews [33,47,50], representing 21 primary studies, included 34,250 participants (85.0%) from racialised backgrounds. However, only one review reported the specific racial/ethnic backgrounds of participants [50], available from all 16 primary studies it included, as: African American/Black/African/Caribbean (22.9%); Asian (including Turkish)/Asian American (29.2%); Hispanic/Latine (1.4%); Irish (5.4%); Surinamese (0.2%); White (excluding Irish) (11.9%); and Other backgrounds (1.0%). Four reviews [32,48,49,51] provided no information on participants' specific racial/ethnic backgrounds from the primary studies they included.

## Exposure definitions

Exposure to racial/ethnic discrimination was defined variously across primary studies: perceived racial/ethnic discrimination (n = 18, 78%) [27,30,31,52–57,59–67]; perceived racial/religious discrimination (n = 3, 13%) [29,68,69]; perceived ethnic discrimination and acceptance (n = 1, 4%) [52], or; not reported (n = 1, 9%) [58]. Various instruments were used to assess racial/ethnic discrimination, the most common of which were self-report questionnaires (author's own or

**Table 2. Characteristics of the primary studies identified by the reviews included in this umbrella review.**

| Variable | Number of reviews reporting[1] | Groups | Number of primary studies | % primary studies |
|---|---|---|---|---|
| **Publication year** | 6 | 1999-2003 | 3 | 13 |
| | | 2004-2008 | 4 | 17 |
| | | 2009-2013 | 6 | 26 |
| | | 2014-2018 | 9 | 39 |
| | | 2019-2023 | 1 | 4 |
| **Study design** | 4 | Cross-sectional | 20 | 87 |
| | | Prospective | 2 | 9 |
| | | Not reported | 1 | 4 |
| **Location of studies** | 4 | Romania | 1 | 4 |
| | | The Netherlands | 5 | 22 |
| | | Norway | 1 | 4 |
| | | UK | 7 | 30 |
| | | US | 8 | 35 |
| | | Not reported | 1 | 4 |
| **RoB assessment tool** | colname="col2">7 | Authors own – results reported | 2 | 4 |
| | | Authors own – results not reported | 6 | 13 |
| | | EPHPP | 16 | 36 |
| | | STROBE checklist | 18 | 40 |
| | | Not Reported | 3 | 7 |
| **Exposure** | 5 | Perceived racial/ethnic discrimination | 18 | 78 |
| | | Perceived racial/religious discrimination | 3 | 13 |
| | | Perceived ethnic discrimination and acceptance | 1 | 4 |
| | | Not reported | 1 | 4 |
| **Exposure instrument(s)[2]** | 2 | EOD | 3 | 13 |
| | | Self-report questionnaire (author's own measure or author details not reported) | 6 | 26 |
| | | Self-report questionnaire (unspecified, not author's own measure) | 7 | 30 |
| | | PRS | 1 | 4 |
| | | Racial Life Events Schedule | 1 | 4 |
| | | EDS | 2 | 9 |
| | | PEDQ-CV | 1 | 4 |
| | | Not reported | 2 | 9 |
| **Exposure frequency[2]** | N/A | Everyday/day-to-day | 3 | 13 |
| | | Past week | 1 | 4 |
| | | Past 3 months | 1 | 4 |
| | | Past year | 1 | 4 |
| | | In current situation (i.e., in current job) | 1 | 4 |
| | | Lifetime | 6 | 26 |
| | | Not specified/Not reported | 13 | 57 |

*(Continued)*

**Table 2.** (Continued)

| Variable | Number of reviews reporting[1] | Groups | Number of primary studies | % primary studies |
|---|---|---|---|---|
| **Type of discrimination measured[2,3]** | N/A | Everyday discrimination | 3 | 13 |
| | | Interpersonal discrimination | 13 | 57 |
| | | Work/school-related discrimination | 8 | 35 |
| | | Institutional discrimination (e.g., related to housing, medical care, loss of job) | 5 | 22 |
| | | Group/vicarious discrimination | 2 | 9 |
| | | Not specified/Not reported | 10 | 43 |
| **Outcomes** | 4 | Psychotic outcomes in clinical samples | 6 | 26 |
| | | Psychotic-like/psychotic experiences in non-clinical samples | 6 | 26 |
| | | Psychotic symptoms in non-clinical samples | 11 | 48 |
| **Outcome instrument(s)[2]** | 2 | BSI | 2 | 9 |
| | | CIS | 1 | 4 |
| | | WHO-CIDI (including version 2.1 and 3.0) | 3 | 13 |
| | | CASH | 1 | 4 |
| | | SCID-I | 2 | 9 |
| | | IRAOS | 1 | 4 |
| | | OCCPI | 1 | 4 |
| | | PS | 1 | 4 |
| | | PAI | 1 | 4 |
| | | PANSS (including SCI-PANSS) | 2 | 9 |
| | | PQ (including full version and the 16-item version, PQ-16) | 6 | 26 |
| | | PSQ | 5 | 22 |
| | | SSPS | 1 | 4 |
| | | Not reported | 2 | 9 |
| **Administration of measure[2,3]** | N/A | Interview/clinician rated (including from case notes) | 11 | 38 |
| | | Self-administrated questionnaire | 16 | 55 |
| | | Not specified/Not reported | 2 | 7 |

Study characteristics of the included 23 primary studies as reported by the included reviews.

[1]The total number of reviews reporting possible was six, as one of the included reviews did not report and provide any data even though it met inclusion criteria based on its intention to include relevant primary studies.

[2]The total percentage for some variables is greater than 100%, as some studies measured exposures and outcomes using more than one instrument or across more than one category.

[3]Not applicable as these variables were generally inferred from relevant exposure/outcome instruments rather than reported by the reviews.

RoB = Risk of bias.

**Exposure measures**: EDS = Everyday Discrimination Scale; EOD = Experiences of Discrimination; PEDQ-CV = Perceived Ethnic Discrimination Questionnaire - Community Version; PRS = Perceived Racism Scale.

**Outcome measures**: BSI = Brief Symptom Inventory; CASH = Comprehensive Assessment of Symptoms and History; CIS = Clinical Interview Schedule; IRAOS = Instrument for Retrospective Assessment of the Onset of Schizophrenia; OCCPI = Operational Criteria for Psychotic Illness; PAI = Personality Assessment Inventory; PANSS = Positive and Negative Syndrome Scale; PQ = Prodromal Questionnaire; PS = Paranoia Scale; PSQ = Psychosis Screening Questionnaire; SCI-PANSS = Structured Clinical Interview for Positive and Negative Syndrome Scale; SCID-I = Structured Clinical Interview for DSM-IV; SSPS = State Social Paranoia Scale; WHO-CIDI = Composite International Diagnostic Interview.

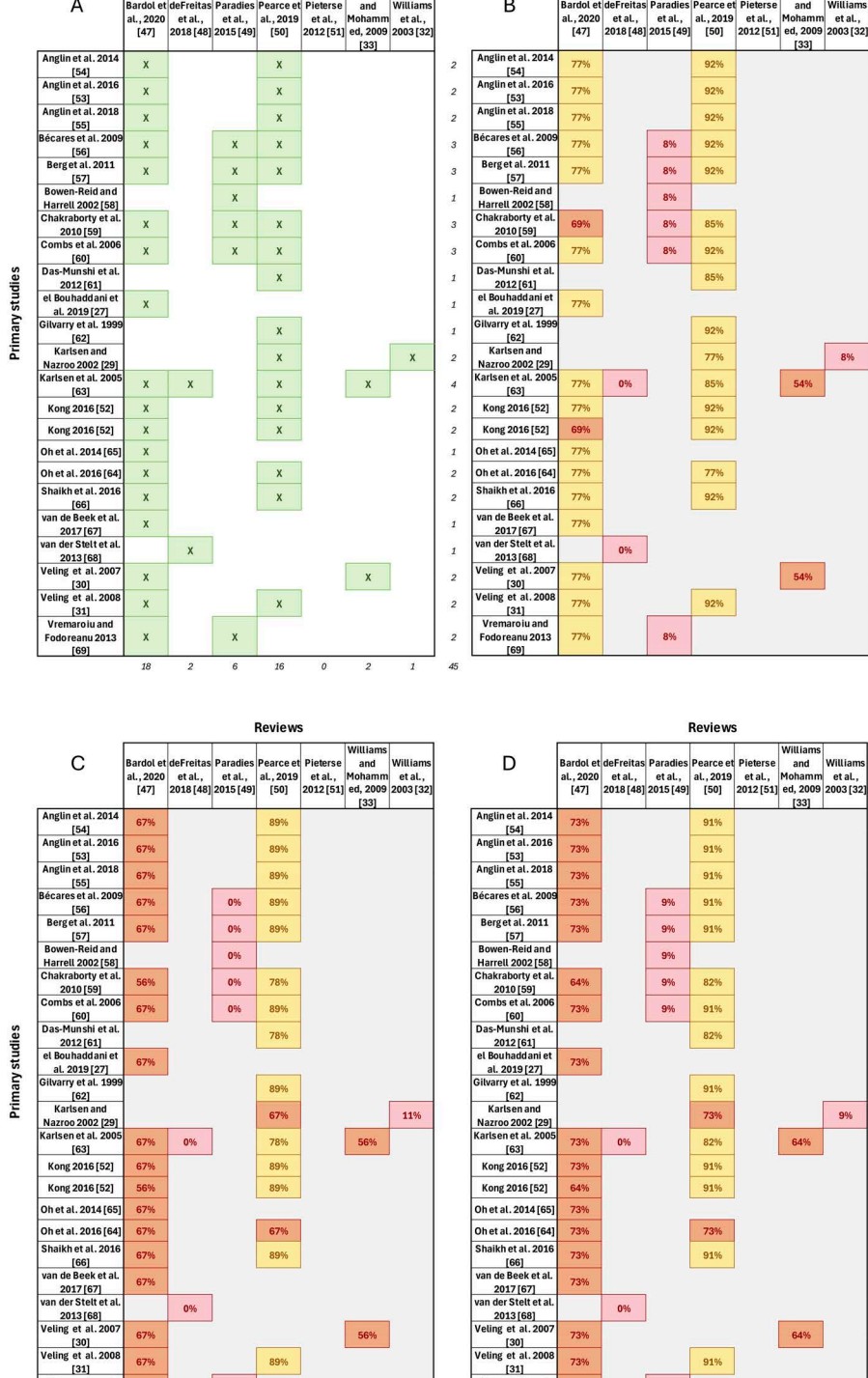

**Fig 2. Citation matrix of overlap in reporting of primary studies by the included reviews.** Citation matrix showing the overlap in reporting of primary studies by the included reviews. **A)** Shows the primary studies reported per review denoted by an 'X'. **B)** Shows the percentage of all characteristics reported within each review (primary study participant characteristics: year of publication, study design, location, risk of bias, total participants, participant age, participant gender, number of participants from a racialised background, participant's racial/ethnic backgrounds; exposure: variable and

instrument; and outcome: variable and instrument) reported within each review. **C)** Provides an overview of reporting of participant characteristics only from each review. **D)** Provides an overview of the reporting of participant characteristics, exposure variable and outcome variable only. For figs B–D: the colours denote the extent of reporting: Red = data for 0–49% characteristics reported; Orange = data for 50–74% characteristics reported; Yellow = data for 75–99% characteristics reported; Green = data for 100% characteristics reported. The percentage reported does not include data provided separately by authors for the purpose of this review. To note, several papers are cited differently between reviews. As such, Anglin [55] is reported as Anglin et al., 2014b in Pearce et al. [50]; Anglin [56] is reported as Anglin et al., 2016 in Pearce et al. [50]; Oh [57] is reported as Oh, 2016 within the primary study characteristics tables in Bardol et al. [47]. Additionally, Kong [52] is shown twice as the paper includes two relevant studies.

**Table 3. Participant characteristics from the primary studies, as reported by reviews included in this umbrella review.**

| | Number of reviews reporting | Groups | Total | Mean | Median | Range (min – max) | % of participants |
|---|---|---|---|---|---|---|---|
| **Number of participants** | 4 | | 40,300 | 1831.8 | 631 | 70–8990 | |
| **Number of participants from racialised backgrounds** | 3 | | 34,250 | 1,427.1 | 295 | 20–8990 | 85.0 |
| **Racial/ethnic background** | 1 | African American/ Black/ African/ Caribbean | 9,220 | 512.3 | 211 | 10–4083 | 22.9 |
| | | Asian (including Turkish)/ Asian American | 11,776 | 512.0 | 642 | 12–1306 | 29.2 |
| | | Hispanic/ Latine[1] | 544 | 135.9 | 156 | 76–156 | 1.4 |
| | | Irish | 2,179 | 726.4 | 728 | 724–728 | 5.4 |
| | | Other | 404 | 44.9 | 20 | 10–100 | 1.0 |
| | | Surinamese | 87 | 28.9 | 32 | 21–34 | 0.2 |
| | | White (excluding Irish) | 4804 | 480.4 | 24 | 2–2975 | 11.9 |
| | | Not Reported | 11,285 | 1612.1 | 267 | 0–8990 | 28.0 |
| **Age (years)** | 1 | | 13,562 | 37.9[2] | | 19.90–44.00 | |
| **Gender** | 1 | Female | 10,209 | 2609.7[2] | 96 | 18–3,834 | 54.7 |
| | | Male | 8,459 | 2246.4[2] | 175 | 32–3,423 | 45.3 |

Participant characteristics of the included 23 primary studies as reported by the included reviews.

'Number of reviews reporting' refers to the number of reviews that reported data for the given characteristic. The total number of possible reporting reviews was six, as one of the included reviews did not report or provide any data.

[1]The original reporting of this category included the gendered term 'latino', which we have reported instead as 'latine' as a preferred, non-gendered term.

[2]These are given as the weighted mean (by study size).

unspecified) (n = 13, 56%) [27,29–31,52,57,60,61,63,67–69] (Table 2). Three studies used the Experiences of Discrimination instrument (13%) [55,56,59], while two primary studies (9%) did not report the measure used [53,58]. Most instruments sought to estimate interpersonal discrimination (n = 13, 57%) [27,30,52,54–56,59,60,62–66]; work/school-related discrimination (n = 8, 35%) [52,54–56,59,62,63,65]; institutional discrimination (e.g., related to housing, medical care, loss of job) (n = 5, 22%) [54–56,59,62], and; group/vicarious discrimination (n = 2, 9%) [27,54]. Most primary studies did not report the frequency over which discrimination was estimated (n = 13, 57%) [27,29–31,52,53,57,58,60,61,63,67–69], but the most commonly reported timeframes included lifetime (n = 6, 26%) [53,55,56,59,62,65] or "everyday" (n = 3, 13%) [62,64,66] discrimination.

## Outcome definitions

The studies also investigated several psychotic outcomes, in both clinical samples (n = 6, 26%) [30,31,54,61,65,67] (including studies of psychotic disorder diagnosis [30,31,54], and studies of symptom severity in ultra-high risk [65] and clinically diagnosed samples [61,67]), as well as non-clinical samples (n = 17, 74%) [27,29,52,53,55–60,62–64,66,68,69] (including studies observing psychotic-like/psychotic experiences and symptoms). Thirteen different psychosis instruments

were used across these studies, or were not reported (n=2, 9%) [53,58], the most common of which were the Prodromal Questionnaire (PQ) (n=6, 26%) [27,55,56,59,65,66] or Psychosis Screening Questionnaire (PSQ) (n=5, 22%) [29,60,63,68,69] (Table 2). Most of the 29 instruments reported were self-administered (n=16, 55%) [27,29,52,55,56,59,60,62,63,65,66,68,69].

### Effects of racial/ethnic discrimination on psychosis

Five reviews explicitly reported the association between racial/ethnic discrimination and psychosis, all finding evidence of positive associations [32,33,47,48,50], including two that provided meta-analytical results [47,48] (Table 4). One meta-analysis reported statistically significant positive associations between racial/ethnic discrimination and psychosis [47], including between *perceived ethnic discrimination* and either *psychotic symptoms* (pooled unadjusted OR (k[number of studies in the meta-analysis]=9) 1.82, 95% CI 1.41, 2.36; pooled adjusted for socio-demographic factors OR 1.77, 95% CI 1.26, 2.49) or *psychotic experiences* (pooled OR (k=7) 1.94, 95% CI 1.42, 2.67), though between-study heterogeneity, as reported, was high ($I^2$=79.08; Q(7)=33.47; p<0.0001). The second meta-analysis also reported a statistically significant correlation between *perceived ethnic discrimination* and *psychotic symptoms* (k=4; r=0.21, 95% CI 0.08, 0.33, z=3.15, p=0.002) [48]; but, as reported, in a sensitivity analysis the authors found that excluding one effect size affected the robustness of the association (r=0.22, z=1.79, p=0.074).

The other two reviews only included psychosis as part of a broader mental health category (e.g., "Other mental health symptoms" including paranoia, psychoticism [49], or "Psychiatric Symptoms (somatic, posttraumatic stress disorder, obsessive-compulsive disorder, psychotic processes, paranoia, but not anxiety or depression)" [51]). The meta-analyses from these reviews reported statistically significant correlations between increased *racial/ethnic discrimination* and worse *mental health symptoms* (k=11, r=-0.21, 95% CI -0.29, -0.12) [49], or *psychiatric symptoms* (k=12, r=0.27, 95% CI 0.20, 0.34) [51]. Furthermore, five of the six relevant primary studies [60–62,67,68] from one of these reviews [49] were also reported by two other included reviews [47,50] (Fig 2) which both did disaggregate data for psychosis and also found evidence of an effect of racial/ethnic discrimination on psychosis.

### Results by outcome

#### Psychotic outcomes in clinical samples

Four reviews [33,47,49,50] reported findings from six primary studies [30,31,54,61,65,67] on the association between racial/ethnic discrimination and psychosis in clinical samples.

Incidence of psychotic disorder. Two studies, both conducted in The Netherlands, investigated incidence of psychotic disorder, with differing results. The first study found a positive association between interpersonal racial/ethnic discrimination and incidence of psychotic disorders in 618 participants [30], and was rated as good quality (by Bardol et al. [47]) (S2 Table). However, a second study found no differences in levels of either perceived interpersonal or vicarious racial/ethnic discrimination in the year prior to onset of schizophrenia compared with controls, in a case-control study with a sample of 263 participants [31]. The study was rated as very good quality by Bardol et al. [47] and as moderate-strong quality by Pearce et al. [50]. A third study found no relationship between perceived racial discrimination and the type of diagnosis or course of illness [54].

Psychotic symptoms. Three studies investigated the association between racial/ethnic discrimination and psychotic symptoms in clinical samples with varied findings [61,65,67]. The first study investigated lifetime perceived racial/ethnic discrimination in a small clinical sample of people with psychotic disorders (n=90) [61], and was rated as very good quality by Bardol et al. [47] but mixed quality by Pearce et al. [50]. It reported a positive correlation with positive (r=0.26, p<0.05), but not negative or cognitive symptoms. A second, smaller (n=70) survey (rated as average quality by Bardol et al. [47]) of Romanian emigrants who had returned to Romania and subsequently been diagnosed with a

**Table 4. Observed associations and meta-analysis results, as reported by reviews included in this umbrella review.**

| Study ID | Number of relevant studies | Relevant exposure variable(s) | Relevant outcome variable(s) | Overall associ-ation | Focus of meta-analysis | k | OR | 95% CI (LL, UL) | Heterogeneity |
|---|---|---|---|---|---|---|---|---|---|
| Bardol et al. [47] | 18 | PED | PS/PEs | Positive association | PED and PS (Pooled unadjusted) | 9 | 1.82 | 1.41, 2.36 | Overall:[1] $I^2 = 79.08$ $Q(7) = 33.47$ $p < 0.0001$ |
| | | | | | PED and PS (Pooled adjusted[2]) | | 1.77 | 1.26, 2.49 | |
| | | | | | PED and PEs (Pooled) | 7 | 1.94 | 1.42, 2.67 | |
| | | | | | Delusional symptoms | 3 | 2.53 | 1.60, 4.01 | $I^2 = 0$ $Q(2) = 0.66$ $p = 0.72$ |
| | | | | | Hallucinatory symptoms | 3 | 1.65 | 1.29, 2.14 | $I^2 = 0$ $Q(2) = 1.08$ $p = 0.58$ |
| de Freitas et al. [48] | 2 | Perception of ethnic discrimination | Manifestation of symptoms of psychosis | Positive association | PED and psychosis symptoms | 4 | $r = 0.21$ $z = 3.15$ $p = 0.002$ | 0.08, 0.33 | $I^2 = 26.77$ $Q = 4.1$ $p = 0.251$ |
| | | | | | Sensitivity analysis | | $r = 0.22$ $z = 1.79$ $p = 0.074$ | | |
| Paradies et al. [49] | 6 | Reported racism | Other mental health symptoms (e.g., paranoia, psychoticism)[3] | Positive association[3] | Reported racism and 'other mental health symptoms' | 11 | $r = -0.2$[4] $z = -4.72$ p-value=<0.001 | -0.29, -0.12 | Q-value = 136.39 p-value Q=<0.001 |
| Pearce et al. [50] | 16 | Perceived racial discrimination | Psychosis/psychosis symptoms | Positive association | N/A | N/A | N/A | N/A | N/A |
| Pieterse et al. [51] | Not reported | Perceived racism | Psychiatric symptoms[5] | Positive association[5] | Racism and Psychiatric Symptoms (mean weighted effect size ($r_{i+}$)) | 12 | 0.27[6] | 0.20, 0.34 | Not reported |
| Williams and Mohammed [33] | 2 | Racial discrimination | Psychosis | Positive association | N/A | N/A | N/A | N/A | N/A |
| Williams et al. [32] | 1 | Perceived racial/ethnic discrimination | Psychosis | Positive association | N/A | N/A | N/A | N/A | N/A |

Summary of the overall results reported by the included reviews. Meta-analytic results are presented as reported.

[1]The heterogeneity has not been ascribed to either of the specific analyses and is instead presented as an overall result as it was unclear from reporting by the review which of the three analyses these results related to.

[2]Additional context for this result was provided by the review: 'Adjusted for socio-demographic factors; n=5'.

[3]The data included in this 'Other mental health symptoms (e.g., paranoia, psychoticism)' outcome did not disaggregate psychosis outcomes from non-psychosis outcomes. Thus, preventing attribution of the results to psychosis alone.

[4]The association was defined by the authors as a statistically significant negative association – increased racism showed greater negative mental health. As this is in alignment of our definition of a positive association (e.g., greater racism related to greater psychosis outcomes), we have referred to it as such in this table.

[5]There were up to 12 studies reported in this psychiatric symptoms outcome. While this category is potentially not relevant for this umbrella review, the author was unable to provide relevant data and therefore, we were unable to determine whether they had identified and included any primary studies on racial/ethnic discrimination and psychosis.

[6]Additional context for this result was provided by the review: 'A shifting unit of analysis approach was used for outcome type. Effect sizes for variable levels not sharing subscripts are statistically significantly different (p <.05), using a Bonferroni-corrected significance level for six comparisons of.008.'

non-affective psychotic disorder [67] found no association between PANSS total symptom severity and perceived racial/ethnic discrimination (r = -0.005, 95%CI -0.240, 0.230, z = -0.041 p = 0.967).

The only study of UHR participants found a positive association between perceived racial/ethnic discrimination and prodromal psychotic symptoms (r = 0.33, p = 0.009), increased levels of perceived racial/ethnic discrimination in the UHR group compared with controls (t = 3.63, p < 0.001), and a positive correlation between perceived racial/ethnic discrimination and persecutory ideation in the whole sample (r = 0.25, p = 0.009) [65]. The study was rated as good quality by Bardol et al. [47] and as mixed quality by Pearce et al. [50].

### Psychotic outcomes in non-clinical samples

Overview. Six reviews [32,33,47–50] reported data relating to psychotic experiences [33,47,48,50] and/or psychotic symptoms [32,47–50] in non-clinical samples. The majority of reviews focused on positive symptoms, including paranoia [47,49,50], unusual thinking [47,50], and altered perceptions (including delusions and hallucinations) [47,50,57]. Two reviews included cognitive disorganisation [47,50] as an outcome. None of the reviews reported data on manic, negative or depressive symptoms in the context of psychosis.

Main review findings. All six reviews found evidence of positive associations between racial/ethnic discrimination and psychotic experiences or symptoms in non-clinical samples, with 13 of 16 primary studies finding statistically significant positive associations [29,52–56,58,60,64,65]. The remaining three studies reported no evidence of an association, observing only non-statistically significant associations [63,68,69]. In their review, Bardol et al. [47] presented meta-analytical evidence for associations between perceived racial/ethnic discrimination and both delusional symptoms (OR (k = 3) 2.53, 95% CI 1.60, 4.01) and hallucinatory symptoms (OR (k = 3) 1.65, 95% CI 1.29, 2.14), as reported in three primary studies [27,57,64]; both meta-analyses showed low heterogeneity (Table 4).

Main primary study findings. The three largest studies [57,60,64] (each with over 4000 participants) all reported statistically significant positive relationships between racial/ethnic discrimination and psychotic experiences (adjusted OR 3.13, p < 0.001) [60] or delusional and hallucinatory symptoms [57,64]. The largest of these studies [64] (n = 8990), also observed a dose-response relationship between greater levels of lifetime perceived racial/ethnic discrimination and increased likelihood of psychotic experiences [64].

Statistically significant positive associations were reported between perceived racial/religious discrimination (OR 1.57, 95% CI 1.02, 2.42) [29] or racial/ethnic discrimination [27,52,53,55–60,62,64] and psychotic experiences or symptoms, including studies that investigated specific outcomes such as delusional symptoms [27,57,64], hallucinatory symptoms [27,57,64], paranoia [52,53,62], schizotypy [58] and attenuated positive psychosis symptoms [55,56,59].

More specifically, one study [62] observed statistically significant associations between racial/ethnic discrimination and levels of paranoia that fell above (r = 0.24, p = 0.008) or below (r = 0.40, p < 0.001) clinical thresholds for disorder in a non-clinical sample of 128 African-American participants. Further, a different study observed statistically significant positive correlations between racial/ethnic discrimination and both paranoid ideation (discrimination in the past year r = 0.25, derived p = 0.001; discrimination in lifetime r = 0.25, derived p = 0.002) and psychoticism (discrimination in the past year r = 0.17, derived p = 0.03; discrimination in lifetime r = 0.16, derived p = 0.05) [53]. Another study investigating the association between perceived racial/ethnic discrimination and schizotypy in a group of 62 Moroccan migrants in The Netherlands, found a statistically significant correlation in participants with a family history of psychopathology (r = 0.51; derived p = 0.01), but not without (r = 0.00) [58]. Additionally, three studies [55,56,59] conducted in the US in a sample of 644 undergraduate students from racialised backgrounds investigated lifetime perceived racial/ethnic discrimination and attenuated positive psychosis symptoms (APPS) and distress associated with these symptoms, two of which reported significant positive correlations related to APPS (r = 0.211, p < 0.001 [56]) and APPS-related distress [55]. The other study [59] also found statistically significant correlations between racial/ethnic discrimination and increased risk/frequency of cognitive disorganisation (r = 0.229/r = 0.234), unusual thinking (r = 0.197/r = 0.204), altered perceptions (r = 0.199/r = 0.196)

and paranoia (r = 0.204/r = 0.210). Greater racial/ethnic discrimination frequency (r = 0.249, p < 0.001) and the number of domains of racial/ethnic discrimination experienced (r = 0.242, p < 0.001) were also correlated with more attenuated positive psychosis symptoms [59].

Mediating and moderating factors. Three reviews [47,49,50] reported findings on the potential mediating and moderating effects of social support, ethnic density, race-based rejection sensitivity, ethnic identity and self-esteem on the association between racial/ethnic discrimination and non-clinical psychotic outcomes (S2 Text). Evidence varied, with two primary studies observing moderating effects of social support [66] and ethnic density [63] on the relationship between racial/ethnic discrimination and psychotic symptoms, but three primary studies finding no such effects for social support [68] or ethnic density [27,60]. Two further primary studies found small, partial mediation of the relationship between lifetime perceived racial/ethnic discrimination and APPS by participants' sensitivity to race-based rejection [55] and ethnic identity [56].

## Results by exposure type

We could not draw overall conclusions on whether psychosis risk differed by type of discrimination experienced, as the reviews generally did not distinguish between type; 57% of reported primary studies investigated interpersonal racial/ethnic discrimination, as inferred from the instruments used (Table 2). Nevertheless, several reviews reported on specific exposures.

Work-related discrimination. Pearce et al. [50] reported a statistically significant relationship between both *interpersonal racism* (OR 2.26, 95% CI 1.62, 3.14, p < 0.001) and *work-related discrimination* (OR 1.46, 95% CI 1.06, 2.00, p = 0.02) and *psychotic experiences* in a UK study of 4281 participants [63]. Additionally, a primary study, from three reviews [47,49,50], reported positive relationships between *verbal insults* and *job refusal*, and *psychotic symptoms* in non-clinical participants [68].

Institutional/structural racism. Two reviews [47,50] reported a US-based study with 4384 participants which observed relationships between potential markers of institutional/structural racism *police abuse*, (adjusted OR 1.69, 95% CI 1.20, 2.39, p < 0.01); *being denied a promotion* (adjusted OR 1.44, 95% CI 1.07, 1.95, p < 0.05); or *being denied a loan* (adjusted OR 1.93, 95% 1.16, 3.26, p < 0.05) and *increased lifetime psychotic experiences* [57].

Verbal and physical attacks. Two primary studies investigated verbal and physical attacks in UK non-clinical samples. The first, from two reviews [32,50], was conducted in a sample of 2507 participants and observed statistically significant associations between both *verbal racial abuse* (OR 2.86, 95% CI 1.69, 4.83) and *physical racial attacks* (OR 4.77, 95% CI 2.32, 9.80), and *psychosis symptoms* [29]. The second study, from four reviews [33,47,48,50], also observed statistically significant associations in 3446 participants between 12-month perceived racial/ethnic discrimination (*verbal abuse* [OR 2.18, 95% CI 1.31, 3.63], and *physical racial attack* [OR 2.94, 95% CI 1.14, 7.57]) and *psychotic experiences* [69].

## Results by primary study characteristics

### Study design

Most studies were cross-sectional (87%); only two studies reported prospective/longitudinal data [52,54]. The first, which was reported by two reviews [47,50], demonstrated a statistically significant relationship between lifetime *perceived racial/ethnic discrimination* and *paranoia* in a non-clinical sample of 116 Asian American employees in the US [52]. The second, reported by one review [50], found no evidence of an association between *perceived racial discrimination* and *type of psychotic disorder diagnosis* (schizophrenia or affective psychosis), or *course of illness* (continuous or episodic) in a sample of 147 participants with a psychotic disorder diagnosis in the UK [54].

### Location of study

No review disaggregated results by country, and our synthesis of the relevant primary studies from these reviews did not identify any systematic differences in the association between racial/ethnic discrimination and any psychosis outcome by country.

### Racial/ethnic background

Five reviews [33,47–50] included investigations of whether the association between racial/ethnic discrimination and psychosis outcomes differed by specific racial/ethnic background, with mixed findings (S3 Text). One review found evidence of an association between perceived *racial/ethnic discrimination* and *psychotic symptoms/experiences* across four of the racial/ethnic groups investigated (Bangladeshi, Black Caribbean, Indian and Pakistani), but no evidence found for the Irish group [47]. Additional studies also reported evidence of positive associations between ethnic/racial discrimination and psychosis outcomes for Black Caribbean [60,61,68,69], Indian [60], and Pakistani [68] groups in studies conducted in the UK [60,68,69] and Norway [61].

### Publication bias

Two reviews found evidence of small study effects or publication bias [48,49], two did not [47,51], and three did not perform quantitative synthesis to permit assessment [32,33,50].

## Discussion

### Principal findings

Systematic reviews found consistent evidence of associations between racial/ethnic discrimination and increased risk of psychotic disorders, symptoms and experiences, as hypothesised. Nonetheless, review quality was low or critically low, sometimes with high levels of heterogeneity in meta-analyses which could impact generalisability and reduce the robustness of findings. Some reviews did not disaggregate findings for psychosis, and we found some evidence of publication bias. Most primary studies were cross-sectional and based on small samples. This suggests that more robust, longitudinal primary studies are required to establish the causal impact of racial/ethnic discrimination on psychosis. Additionally, no studies were conducted outside of Europe or the US, raising issues of generalisability.

In general, the evidence base was larger and more consistent for psychotic experiences/symptoms in non-clinical samples than in clinical samples, although both provided evidence of positive associations. The largest effect sizes reported (correlation coefficients and odds ratios, respectively) were observed in non-clinical settings between perceived racial/ethnic discrimination and schizotypy in participants with a family history of psychopathology ($r = 0.51$; derived $p = 0.01$) [58] and between physical racial attacks and psychosis symptoms (OR 4.77, 95% CI 2.32, 9.80) [29]. Several other studies also observed large odds ratios of at least 2.00 [60,63,69]. The largest effect size reported in the clinical setting was observed between perceived racial/ethnic discrimination and prodromal psychotic symptoms in UHR participants ($r = 0.33$, $p = 0.009$) [65]. In the limited data available, patterns of positive associations were also evident for several racialised groups independently (Bangladeshi, Black Caribbean, Indian and Pakistani). There were insufficient data to answer the remaining questions around impact of racial/ethnic discrimination type, country, and study design, on the association.

### Limitations

First, our review may have amplified publication bias, by including reviews and primary studies with risk or evidence of such bias. Nevertheless, the applicability of the AMSTAR-2 tool for non-intervention focused reviews has been questioned [41], with high proportions of reviews based on observational data consistently assessed as low quality under AMSTAR-2, despite contrary evidence of their robustness [67,70]. For this reason, we also calculated the percentage of total AMSTAR-2 items met by each to provide a potentially more nuanced overview of each review's quality. We also note that two reviews [31,32] were undertaken before the development of PRISMA guidelines, meaning that AMSTAR-2 criteria were applied retrospectively. Additionally, we were unable to directly assess small study effects due to the lack of sufficient data to conduct our own meta-analyses.

Second, as we followed best practice in umbrella reviewing – to report only data from the reviews themselves – this also introduced the potential to have exacerbated errors or misreporting in the original reviews, and thus the potential to miss or incorrectly report findings.

Third, although we comprehensively searched four major databases (Medline, Embase, PsycINFO, ProQuest Central), our additional search of Google Scholar was limited to the first 400 records (50 from each search concept) due to resourcing constraints (see S1 Text). While there is a small possibility we missed unpublished reviews, reviews only indexed in Google Scholar, or those in the grey literature, our backward- and forward- citation searches would have been very likely to alert us to any relevant extant reviews we had missed in our original search; this process led to the identification of two published reviews we included in this umbrella review, but no relevant grey literature, suggesting the impact of omitting the grey literature in our study design would have been small.

Finally, it was beyond the scope of our review to include reviews of broader structural factors (i.e., income, education, housing, poverty) and psychosis that may themselves be determined by systemic and institutional forms of racism; there is increasing evidence that these racially-determined social inequalities could explain much of the excess risk in psychosis [20]. Indeed, the public mental health impact of systemic forms of racism on psychosis may be just as substantial as the pernicious impact that interpersonal racism plays in the aetiology of psychosis [21,70,71]. We therefore suggest our findings are likely to be an under-representation of the total effect of all forms of racism on psychosis.

## Evidence in context and implications

The findings from our review indicate a relationship between racial/ethnic discrimination and psychosis. This potentially supports theories about the pernicious role that racism plays in explaining a substantial portion of the large ethnic disparities observed in psychosis risk that have been consistently observed in many marginalised ethnic and migrant groups for over 170 years [72]. Despite this, our review reveals that much of the current evidence, when systematically appraised, suffers from substantive methodological issues, and possible publication bias. We also noted that the field is occasionally hampered by problems of nomenclature; for example, a previous umbrella review purported to investigate the association between racial/ethnic discrimination and psychosis in "Black people and people of color", but whose participants included only migrants [73]. This may be a legacy of migration-focused research in the field, but the narrative language we adopt must reflect and carefully disentangle disparate experiences of migration and racial/ethnic marginalisation.

Our systematic review evidence is supported by many non-systematic reviews [14,19,21,37,70,74–78] outside the scope of this review. Earlier selective reviews often considered racial/ethnic discrimination alongside the other main hypotheses for excess psychosis risk amongst migrant and racialised ethnic groups [14,37,74,78]. Here, experiences of racism or racial/ethnic discrimination were contextualised alongside other social determinants of health, including lower socioeconomic status and neighbourhood-level social inequalities, potential examples of the aforementioned structural and systematic forms of racism in housing, education, employment and income domains that may exacerbate inequalities in psychosis outcomes by race and ethnicity [37,74]. More recent selected reviews have focussed more explicitly on the potential role for various forms of racial/ethnic discrimination, including structural racism, to account for higher levels of psychosis in some racialised ethnic groups [19,21,70], particularly in the US context [19,21].

Our findings also cohere with the wider literature that supports an association between different forms of racial/ethnic discrimination and other mental health outcomes, especially depression and anxiety [24,79–82]. Four reviews included in this umbrella review also reported evidence of positive associations between racial/ethnic discrimination and depression (major depression or symptoms) [32,33,48,49], anxiety (generalised anxiety disorder or symptoms) [32,33,49] and poorer overall mental health [32,33,48,49].

Together, these findings add to our wider understanding of the deleterious and persistent impact of racism on mental health. Yet, racism is a putatively modifiable risk factor, and thus we can investigate targetable points on which to intervene to reduce inequities in mental ill health. There is already evidence that the geopolitical environment can play a role

in exacerbating racial injustice and disparities in mental health [19,82], however, these same institutions and policymakers are also instrumental if we wish to drive equitable public mental health, where socially-just policymaking in education, housing, employment and health will be fundamental to improving mental health in racialised groups.

Our findings have potential implications for clinical practice. As psychosis has a young median age of onset (25 years old) [83] and a younger age for psychotic-like experiences [84], psychosocial interventions that help people potentially exposed to discrimination could be focused on children and adolescents to foster social support and collective self-esteem, which have been shown to decrease psychosis risk in those who have experienced racism [52,66,85]. Adolescents and adults exposed to racism should continue to receive these interventions after onset and diagnosis, including approaches that target the impact of racism through education and racial stress/trauma-focused CBT [19]. As part of a holistic approach, such reform would need to be accompanied by evidence-based anti-racism training to educate clinical staff on the impacts of racism on health and improve their competencies in discussing racism experiences, as well as to prevent them re-traumatising or further exposing those receiving care, all of which also benefit the organisations providing care [86].

Interventions should also be holistic, both targeting systems that generate racism to address the structural conditions that permit experiences of racism to perpetuate, and to provide support to individuals to help mitigate the negative psychological impacts of racism. An example of a new national intervention from the UK is the Patient and carer race equality (PCREF) framework [87], which has been developed in collaboration with patients, carers and community organisations and introduced across NHS England Trusts. This anti-racism framework aims to provide Trusts with the tools to interrogate and tackle racism within their practices and policies to address the current racial/ethnic inequalities. Interventions could also look to support culturally- and community- grounded early intervention for those experiencing or at risk of experiencing racial/ethnic discrimination, such as the programmes described and discussed by Jones and Neblett [88]. Since experiences of racial/ethnic discrimination can differ by country and both between and within racialised groups [21,89–92], such interventions should provide specific support relating to their experiences. Additionally, we need more research investment to understand the most appropriate intersectional interventions to prevent mental ill health in people who face discrimination as a result of multiple intersecting marginalised identities, including by sexuality, gender, and religion [92].

## Recommendations for future research

Discussions and education around racism should not be limited to clinical practice but are a vital issue for researchers in a field that occupies the nexus of medicine, science and the social sciences. Conversations about racism can still be fraught with disbelief and denial [93,94], and while this review presents findings that should further help combat this, a recent review demonstrated the continued issue of racial bias within scientific publishing itself [95]. Therefore, continued publication by journals and academics on the impact of racism is also integral to driving understanding, education and change.

Beyond this, the evidence in this review calls for more investment in rigorous, longitudinal research in diverse populations and settings, as well as stronger adherence to reporting guidelines for both primary studies and systematic reviews. Longitudinal study designs may help us to elucidate whether there are specific times in the life course when experiences of racial/ethnic discrimination are most impactful, as well as to explore the mechanisms underlying the association. However, racism is a pervasive yet personal experience which cannot be wholly captured through longitudinal studies, and explorations outside of such rigid methodologies may be better suited to understanding the full reality and human experience of racism, and therefore its impacts on mental health.

Several studies and reviews have posited that mechanisms such as allostatic load [96–98], altered threat processing [99–101] or changes to social cognition [20,102] could potentially link racism neurobiologically to various mental health outcomes and should be explored. Additionally, further exploration into potential protective factors – such as ethnic density, racial identity, racial socialisation and cultural worldview [19,88] – will also be important in elucidating further putative prevention strategies to tackle ethnic and racial inequalities in mental health.

## Conclusion

This umbrella review provides consistent evidence of associations between racial/ethnic discrimination and psychosis in both clinical and non-clinical populations. There was a greater evidence base for the latter with 13 of the 16 primary studies included observing statistically significant positive associations and several of those reporting large odds ratios of at least 2.00 [29,60,63,69]. These findings may help to explain the excess risk seen in racialised groups in many countries across Europe [28] as well as in the US, Canada and Australia.

Despite this, we observed notable issues in the extant literature, including methodological heterogeneity, review quality and a focus on interpersonal racism over other forms including structural and institutional racism. We recommend the need for high quality, longitudinal research that explores the mechanisms and interventions that, respectively, could produce and prevent excess psychosis risk that arise after pervasive exposure to multiple, ongoing forms of racism in society. The findings of this umbrella review point towards a role for holistic approaches to clinical practice which include targeted interventions for those who may experience racial/ethnic discrimination, education for clinical staff on the impact of racism and how to avoid re-traumatising those receiving care, as well as interventions that target the systems that uphold and enable racism. Future research should also aim to identify protective factors which can be integrated into clinical interventions and policy to improve people's mental wellbeing and tackle racism.

## Supporting information

**S1 Checklist.**
(DOCX)

**S1 Text. Umbrella Review Search Strategy.**
(DOCX)

**S2 Text. Additional Results: Mediators/Moderators of the association between racial/ethnic discrimination and psychosis outcomes in non-clinical samples.**
(DOCX)

**S3 Text. Additional Results: Results by Racial/Ethnic Background.**
(DOCX)

**S1 Table. AMSTAR-2 Appraisal.** Results of the AMSTAR-2 risk of bias assessment and overview of the total AMSTAR-2 items met by each review.
(DOCX)

**S2 Table. Primary Studies Risk of Bias.** Summary of risk of bias assessments and scores of the included primary studies, as conducted and reported by the reviews within which they were reported.
(DOCX)

## Acknowledgments

The authors would like to acknowledge the work of Dr Debora Marletta, Training and Clinical Support Librarian at UCL Library Services, who supported with the systematic search strategy.

## Author contributions

**Conceptualization:** India Francis-Crossley, Lasana Harris, Juliana Onwumere, James B. Kirkbride.

**Data curation:** India Francis-Crossley, Georgie Hudson.

**Investigation:** India Francis-Crossley, Georgie Hudson.

**Methodology:** India Francis-Crossley, Lasana Harris, Juliana Onwumere, James B. Kirkbride.

**Project administration:** India Francis-Crossley.

**Supervision:** Juliana Onwumere, James B. Kirkbride.

**Validation:** India Francis-Crossley, Georgie Hudson, James B. Kirkbride.

**Visualization:** India Francis-Crossley.

**Writing – original draft:** India Francis-Crossley.

**Writing – review & editing:** India Francis-Crossley, Georgie Hudson, Lasana Harris, Juliana Onwumere, James B. Kirkbride.

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
