## [Decision Letter · Decision Letter 0]

7 May 2025

PMEN-D-25-00026

The association between racism and psychosis: An umbrella review

PLOS Mental Health

Dear Dr. Francis-Crossley,

Thank you for submitting your manuscript to PLOS Mental Health. After careful consideration, we feel that it has merit but does not fully meet PLOS Mental Health’s publication criteria as it currently stands. Therefore, we invite you to submit a revised version of the manuscript that addresses the points raised during the review process.

We look forward to receiving your revised manuscript.

Kind regards,

Juan Felipe Cardona, Ph.D.

Academic Editor

PLOS Mental Health

Journal Requirements:

1. We ask that a manuscript source file is provided at Revision. Please upload your manuscript file as a .doc, .docx, .rtf or .tex.

Additional Editor Comments (if provided):

Reviewers' comments:

Reviewer's Responses to Questions

**Comments to the Author**

1. Does this manuscript meet PLOS Mental Health’s publication criteria?

Reviewer #1: Yes

Reviewer #2: Partly

2. Has the statistical analysis been performed appropriately and rigorously?

Reviewer #1: Yes

Reviewer #2: Yes

3. Have the authors made all data underlying the findings in their manuscript fully available (please refer to the Data Availability Statement at the start of the manuscript PDF file)?

Reviewer #1: Yes

Reviewer #2: Yes

4. Is the manuscript presented in an intelligible fashion and written in standard English?

Reviewer #1: Yes

Reviewer #2: Yes

Reviewer #1: Introduction:

• Line 69 could benefit from further explaining the specific adverse outcomes of people with psychosis

• Some inconsistencies in the way statistics are reported (numeric vs spelled out)

• Lines 95 – 103 are not especially clear in the flow of explaining the differing levels of racism. There could be edits for overall logical flow.

Methods:

• It would be beneficial to further explain the adaptation in Google Scholar that was used for searching

Overall:

• Highlighting the most correlated causal results could make the results and discussion stronger. Authors could consider a figure to highlight the relationships.

Reviewer #2: Peer Review of "The Association Between Racism and Psychosis: An Umbrella Review"

Methodology

The authors conducted a comprehensive umbrella review, systematically searching multiple databases (Medline, Embase, PsycINFO, ProQuest Central, and Google Scholar) for systematic reviews and meta-analyses related to racial/ethnic discrimination and psychosis. The methodology is well-structured, adhering to PRISMA guidelines and pre-registering the protocol (CRD42023400656).

Key methodological strengths include:

Rigorous Inclusion Criteria: The review included systematic reviews and meta-analyses that specifically investigated non-organic psychosis and racial/ethnic discrimination, ensuring a focused analysis.

Risk of Bias Assessment: The authors employed the AMSTAR-2 tool to assess the quality of included reviews, which is a recognized method for evaluating systematic reviews.

Data Extraction and Synthesis: Data were extracted narratively, and the authors attempted meta-analyses where possible, although they faced challenges due to heterogeneity among studies.

However, there are some limitations:

High Heterogeneity: The review noted significant heterogeneity in the studies, which complicates the synthesis of results and limits the generalizability of findings.

Quality of Included Reviews: Most reviews were rated as low or critically low quality, which raises concerns about the reliability of the conclusions drawn from the synthesized evidence.

Clarity

The manuscript is generally well-written and organized, with a clear structure that guides the reader through the introduction, methodology, results, and discussion. The use of headings and subheadings enhances readability.

Strengths in clarity include:

Clear Definitions: The authors provide clear definitions of key terms, such as "racial/ethnic discrimination" and "psychosis," which helps contextualize the research.

Logical Flow: The progression from the introduction to the results and discussion is logical, making it easy for readers to follow the authors' arguments.

Areas for improvement:

Complexity of Language: Some sections could benefit from simpler language or clearer explanations, particularly when discussing statistical analyses and methodological terms.

Figures and Tables: The inclusion of figures (e.g., PRISMA diagram) and tables summarizing key findings could enhance understanding, but they should be clearly labeled and referenced in the text.

Results

The results of the umbrella review indicate a consistent positive association between racial/ethnic discrimination and psychosis, with all included reviews finding evidence supporting this relationship. Notably:

Quantitative Findings: The meta-analyses reported significant odds ratios for psychotic symptoms and experiences, suggesting a robust link between discrimination and psychosis.

Population Differences: The authors observed stronger evidence for psychotic outcomes in non-clinical populations compared to clinical samples, which is an important distinction.

However, the results are tempered by:

Quality of Evidence: The overall quality of the evidence is low, with many studies being cross-sectional and suffering from methodological limitations.

Need for Further Research: The authors emphasize the necessity for high-quality longitudinal studies to better understand the causal pathways between racial/ethnic discrimination and psychosis.

Conclusion

Overall, the manuscript presents a valuable synthesis of the existing literature on the relationship between racism and psychosis. While the methodology is sound and the results are significant, the low quality of the included studies and high heterogeneity highlight the need for caution in interpreting the findings. Future research should aim to address these gaps, focusing on high-quality longitudinal studies to clarify the mechanisms at play. The clarity of the writing is generally good, but some sections could be simplified for broader accessibility.

**Do you want your identity to be public for this peer review?** For information about this choice, including consent withdrawal, please see our Privacy Policy

Reviewer #1: No

Reviewer #2: No

---

## [Decision Letter · Decision Letter 1]

23 Jun 2025

PMEN-D-25-00026R1

The association between racism and psychosis: An umbrella review

PLOS Mental Health

Dear Dr. Francis-Crossley,

Thank you for submitting your manuscript to PLOS Mental Health. After careful consideration, we feel that it has merit but does not fully meet PLOS Mental Health’s publication criteria as it currently stands. Therefore, we invite you to submit a revised version of the manuscript that addresses the points raised during the review process.

EDITOR: Please insert comments here and delete this placeholder text when finished. Be sure to:

Indicate which changes you require for acceptance versus which changes you recommendAddress any conflicts between the reviews so that it's clear which advice the authors should followProvide specific feedback from your evaluation of the manuscript

Please ensure that your decision is justified on PLOS Mental Health’s publication criteria  and not, for example, on novelty or perceived impact.

We look forward to receiving your revised manuscript.

Kind regards,

Juan Felipe Cardona, Ph.D.

Academic Editor

PLOS Mental Health

Journal Requirements:

1. As required by our policy on Data Availability, please ensure your manuscript or supplementary information includes the following:

Additional Editor Comments (if provided):

Reviewers' comments:

Reviewer's Responses to Questions

**Comments to the Author**

Reviewer #1: All comments have been addressed

Reviewer #3: All comments have been addressed

publication criteria?

Reviewer #1: Yes

Reviewer #3: Yes

3. Has the statistical analysis been performed appropriately and rigorously?

Reviewer #1: Yes

Reviewer #3: Yes

4. Have the authors made all data underlying the findings in their manuscript fully available (please refer to the Data Availability Statement at the start of the manuscript PDF file)?

Reviewer #1: Yes

Reviewer #3: Yes

5. Is the manuscript presented in an intelligible fashion and written in standard English?

Reviewer #1: Yes

Reviewer #3: Yes

Reviewer #1: Thank you for the opportunity to review this important and timely manuscript. This paper provides a much-needed synthesis of existing systematic reviews on a critically understudied area in mental health research. The topic is highly relevant, the methodology is rigorous, and the findings contribute meaningfully to the field. I commend the authors on their transparent reporting, use of a pre-registered protocol, and thorough adherence to PRISMA and Cochrane guidance.

Strengths:

The study addresses a pressing and understudied issue: the role of racial/ethnic discrimination in psychosis.

Methodological rigor is evident in the comprehensive search strategy, dual screening, and use of AMSTAR-2.

The narrative synthesis is clear, well-structured, and appropriately cautious given the heterogeneity and quality of included reviews.

Meta-analytic findings are presented with nuance, and the use of both clinical and non-clinical samples is a particular strength.

Terminological choices (e.g., “racialised backgrounds,” “Latine”) demonstrate cultural sensitivity and reflexivity.

Suggestions for Minor Revision:

Interpretation of Heterogeneity:

Given the high heterogeneity in several reported meta-analyses (e.g., I² = 79%), I recommend including a brief interpretive discussion of how this may affect the robustness or generalizability of the findings.

Terminology Clarification:

The authors adopt inclusive and up-to-date terminology throughout (e.g., avoiding BAME). It may be helpful to briefly address how historical terminology in older reviews was handled or interpreted, especially when terminology was inconsistent or outdated.

Expanded Implications:

The discussion on public health implications is well-stated, but could benefit from a slightly more detailed outline of potential intervention targets (such as structural vs interpersonal interventions, implications for health services, or culturally grounded early intervention models).

Proofreading:

A few minor typographic and formatting issues (e.g., spacing inconsistencies, capitalization) remain and can be addressed in a final edit.

Reviewer #3: General comments and some key concerns:

Dear authors, thank you for making the revision and responding to the comments as a way to improve on the manuscript titled “The association between racism and psychosis: An umbrella review”. However, the following are some few issues highlighted that need further attention.

1. Methodology

The authors should use “Methodology” instead of “Materials and Methods”.

2. Results

The authors should include the flow chart of how papers included in the study were arrived at. The authors should present the percentages to 1 decimal place in the whole document and they should be reported as %(n) and not n(%). In table 2 and 3, the percentages should be removed from each value since it is already included in the sub-title in the column. However, this was not affected from the previous comments to the authors. What is the unit s of age in table 3 and it should be written as age (years or months or weeks depending on the interest)? The results section needs to be rearranged so that it can flow.

3. Conclusion

The authors should include the conclusion based on the findings from the study.

**Do you want your identity to be public for this peer review?** For information about this choice, including consent withdrawal, please see our Privacy Policy

Reviewer #1: No

Reviewer #3: No

---

## [Editor Report · Decision Letter 2]

10 Jul 2025

PMEN-D-25-00026R2

The association between racism and psychosis: An umbrella review

PLOS Mental Health

Dear Dr. Francis-Crossley,

Thank you for submitting your manuscript to PLOS Mental Health. After careful consideration, we feel that it has merit but does not fully meet PLOS Mental Health’s publication criteria as it currently stands. Therefore, we invite you to submit a revised version of the manuscript that addresses the points raised during the review process.

We look forward to receiving your revised manuscript.

Kind regards,

Juan Felipe Cardona, Ph.D.

Academic Editor

PLOS Mental Health

Journal Requirements:

Additional Editor Comments (if provided):

Dear Dr. Francis-Crossley and colleagues,

Thank you once again for your thoughtful resubmission of the manuscript. I appreciate the sustained effort that you and your co-authors have dedicated to addressing the reviewers’ feedback across rounds. Please accept my apologies for the delay in response. The evaluation process was prolonged due to reviewer availability and internal considerations. We appreciate your patience and continued commitment to the process.After a careful assessment of the revised version and based on the comments from Reviewer 1, I am pleased to report that your manuscript is very close to being ready for acceptance. However, we would kindly ask you to consider a small number of minor editorial revisions, which will help ensure clarity and consistency in the final version.

Specifically, expand the concluding paragraph to better summarise the review’s key findings and implications for policy and future research, provide a brief clarification on the exclusion of grey literature in your search strategy, and how this might impact the review's comprehensiveness or introduce potential bias?. Include a concise summary of the types of psychotic outcomes such as clinical diagnosis, subclinical symptoms, considered across included studies to enhance the interpretability of your synthesis.

These refinements are intended to reinforce the clarity and impact of your manuscript.

Importantly, no further responses are needed to Reviewer 2, and we ask you to disregard those comments at this stage.

Warm regards,

Juan Felipe Cardona, Ph.D.

Academic Editor

PLOS Mental Health
---

## [Editor Report · Decision Letter 3]

22 Jul 2025

The association between racism and psychosis: An umbrella review

PMEN-D-25-00026R3

Dear Ms Francis-Crossley,

We are pleased to inform you that your manuscript 'The association between racism and psychosis: An umbrella review' has been provisionally accepted for publication in PLOS Mental Health.

Best regards,

Juan Felipe Cardona, Ph.D.

Academic Editor

PLOS Mental Health

Dear Dr. Francis-Crossley

Thank you for your responses and the revised submission. I have reviewed the changes and confirm that the manuscript now meets the editorial and methodological requirements for publication.

Best regards,

Juan Felipe Cardona Londoño, Ph.D.

Academic Editor

PLOS Mental Health